

# Higher bee abundance, but not pest abundance, in landscapes with more agriculture on a late-flowering legume crop in tropical smallholder farms

Cassandra Vogel[1], Timothy L. Chunga[2], Xiaoxuan Sun[3], Katja Poveda[4] and Ingolf Steffan-Dewenter[1]

[1] Department of Animal Ecology and Tropical Biology, Biocenter, University of Würzburg, Würzburg, Germany
[2] Soils, Food and Healthy Communities, Ekwendeni, Mzimba District, Malawi
[3] Department of Geography, University of Western Ontario, London, Ontario, Canada
[4] Department of Entomology, Cornell University, Ithaca, New York, United States of America

Corresponding author
Cassandra Vogel,
cassandra.vogel@uni-wuerzburg.de

## ABSTRACT

**Background**. Landscape composition is known to affect both beneficial insect and pest communities on crop fields. Landscape composition therefore can impact ecosystem (dis)services provided by insects to crops. Though landscape effects on ecosystem service providers have been studied in large-scale agriculture in temperate regions, there is a lack of representation of tropical smallholder agriculture within this field of study, especially in sub-Sahara Africa. Legume crops can provide important food security and soil improvement benefits to vulnerable agriculturalists. However, legumes are dependent on pollinating insects, particularly bees (Hymenoptera: Apiformes) for production and are vulnerable to pests. We selected 10 pigeon pea (Fabaceae: *Cajunus cajan* (L.)) fields in Malawi with varying proportions of semi-natural habitat and agricultural area within a 1 km radius to study: (1) how the proportion of semi-natural habitat and agricultural area affects the abundance and richness of bees and abundance of florivorous blister beetles (Coleoptera: *Melloidae*), (2) if the proportion of flowers damaged and fruit set difference between open and bagged flowers are correlated with the proportion of semi-natural habitat or agricultural area and (3) if pigeon pea fruit set difference between open and bagged flowers in these landscapes was constrained by pest damage or improved by bee visitation.
**Methods**. We performed three, ten-minute, 15 m, transects per field to assess blister beetle abundance and bee abundance and richness. Bees were captured and identified to (morpho)species. We assessed the proportion of flowers damaged by beetles during the flowering period. We performed a pollinator and pest exclusion experiment on 15 plants per field to assess whether fruit set was pollinator limited or constrained by pests.
**Results**. In our study, bee abundance was higher in areas with proportionally more agricultural area surrounding the fields. This effect was mostly driven by an increase in honeybees. Bee richness and beetle abundances were not affected by landscape characteristics, nor was flower damage or fruit set difference between bagged and open flowers. We did not observe a positive effect of bee density or richness, nor a negative effect of florivory, on fruit set difference.

**Discussion**. In our study area, pigeon pea flowers relatively late—well into the dry season. This could explain why we observe higher densities of bees in areas dominated by agriculture rather than in areas with more semi-natural habitat where resources for bees during this time of the year are scarce. Therefore, late flowering legumes may be an important food resource for bees during a period of scarcity in the seasonal tropics. The differences in patterns between our study and those conducted in temperate regions highlight the need for landscape-scale studies in areas outside the temperate region.

## INTRODUCTION

Despite covering 16% of global agricultural area and supporting some of the world's most vulnerable populations, tropical smallholder agriculture has received relatively little attention in agroecological research compared to large-scale agriculture in temperate regions (*Steward et al., 2014*). Smallholder agriculture often exists within the world's most biodiverse but also threatened landscapes, creating a necessity to develop smallholder agriculture in sustainable ways that can both improve food security, whilst also safeguarding biodiversity and ecosystem functions (*Newbold et al., 2015*; *Samberg et al., 2016*).

Insect communities in agricultural fields are driven by field management and the composition of the surrounding landscape (*Martin et al., 2019*). These insect communities are known to be of large importance to agricultural productivity. Pollination, particularly by bees, is a key ecosystem service that is essential for enhancing the production of fruits and seeds in a majority of crops (*Garibaldi et al., 2013*). On the other hand, herbivory of crops by insect pests is estimated to cause more than 10% of pre-harvest losses (*Oerke, 2006*). A recent meta-analysis has shown that pollinator density and richness benefits from a more complex landscape containing more semi-natural habitat (SNH) (*Dainese et al., 2019*). For insect pests, this pattern is more inconsistent between studies than for pollinators (*Karp et al., 2018*). Some studies show decreasing pest pressure with increasing semi-natural habitat, often attributed to increased occurrence of natural enemies in landscapes with more semi-natural habitat (*Chaplin-Kramer & Kremen, 2012*). Others suggest that semi-natural habitats can be a source of pests for crops (*Rusch et al., 2013*) as, for example, non-crop habitat can be a refuge in which insect pests can survive outside of the growing season, only to recolonize crops once they start growing again (*Bianchi, Booij & Tscharntke, 2006*; *Martin et al., 2019*).

Despite being well studied in temperate larger-scale agriculture, larger knowledge gaps still exist on the understanding of landscape effects on beneficial and damaging insects in tropical smallholder agriculture, particularly in Africa (*Otieno et al., 2020*). Even if landscape-scale studies in Africa are conducted, they usually focus on more commercially important crops, such as coffee and cotton (*Vanlauwe et al., 2014*). Crops more important

to household food security are understudied in comparison, despite Africa's high rates of food insecurity (*Sasson, 2012*; *Graeub et al., 2016*). Food insecurity in Africa is caused in part by large crop losses due to pests, with farmers having limited access to pest management strategies, such as pesticides (*Abate, Van Huis & Ampofo, 2000*). Though pesticide use has increased in Africa in the last decades, pesticide application may not necessarily reduce crop losses by pests despite significant costs to the environment and to human health, particularly in sub-Saharan Africa, where lack of access to safety equipment and knowledge on how to correctly apply pesticides increases personal health risks to farmers and reduces the potential pest-control benefits (*Oerke, 2006*; *De Bon et al., 2014*; *Isgren & Andersson, 2020*)). This further highlights the need to understand what drives pest densities on important crops in the region in order to successfully manage them sustainably (*De Bon et al., 2014*). Particularly, legume crops are an important addition to cereal staple crops for providing food security and nutrition in sub-Saharan Africa (*Otieno et al., 2020*).

Pigeon pea (Fabaceae: *Cajunus cajan* (L.)) is a legume crop with the potential to improve livelihoods of smallholder farmers due to its unique combination of high nutritional value, drought tolerance and nitrogen-fixing, soil-improving properties (*Odeny, 2007*). However, the adoption of pigeon pea in our study area of northern Malawi has been constrained by perceived yield losses by farmers due to a flower-feeding blister beetle (Coleoptera: *Melloidae*) (*Mhango, Snapp & Phiri, 2013*). The most commonly observed blister beetle on pigeon pea is a *Hycleus* species (Appendix 1), which often feeds on the entire flower, including the reproductive parts. The damaged flower, therefore, is unable to set fruit and produce any yield. *Hycleus* sp. is common pest on legume crops in Africa (*Lebesa et al., 2012*). Average production in Malawi, one of the larger pigeon pea growing regions in Africa, is less than a quarter of potential yields (*Odeny, 2007*). In general, yield losses of pigeon pea due to insect pests range from 10–70% (*Otieno et al., 2020*), though the blister beetles are viewed as the most constraining to yield (*Mhango, Snapp & Phiri, 2013*). Pigeon pea can be up to 70% self-pollinating (*Saxena, Singh & Gupta, 1990*). However, pollination has been shown to significantly improve fruit set of pigeon pea compared to unvisited flowers. In particular, bees of the genera *Megachile* and *Xylocopa* have been found to be responsible for 20–90% of cross-pollination in this crop, with the remainder being pollinated by other bee species or pollinating flies (*Fohouo, Pando & Tamesse, 2014*; *Otieno et al., 2015*; *Otieno et al., 2020*).

We investigated how the proportion of semi-natural habitat and agricultural area within a 1 km radius around ten pigeon pea fields affects (1) the abundance and species richness of bees (Hymenoptera: Apiformes) and the abundance of florivorous blister beetles, and (2) if the proportion of flowers damaged and fruit set difference is correlated with the proportion of agricultural area or semi-natural habitat. Additionally, using an exclusion experiment, we (3) investigated if differences in fruit set between visited (open) and unvisited (bagged) flowers set in these landscapes were constrained by pest damage and or improved by bee visitation.

## MATERIALS & METHODS

### Study area and field selection

We conducted our study from May to August 2019 in Mzimba district, Northern Malawi. We selected ten already existing pigeon pea fields. We were granted verbal permission for conducting the research on each of the farmer's private fields. Their names are: Isobel Lubanda, Adams Tembo, Mercillina Tembo, Ireen Mhoni, Simon Chitaya, Jacob Mvula,Jane Salanda, Lyna Njunga, Goodson Moyo, Moles Thupa. The farmers are not represented by a company or a farming cooperation, but were in contact with the authors through the SFHC (Soils, Food and Healthy Communities) organization. We have no form of written permission for the conduction of the research. In all the ten fields, the pigeon pea crop had been planted at the onset of rains in December 2018, and were initially intercropped with groundnut (Fabaceae: *Arachis hypogaea* L.). By the time we began data collection, all the groundnut had already been harvested from all the fields. All pigeon pea fields selected were planted with a local medium-maturing variety. The peak of bloom of this pigeon pea variety is in May in our system. The duration of the flowering period can depend on the climatic conditions, but in our region, the bloom lasted about 4 weeks.

Malawi is located in the seasonal tropics and experiences a marked peak in rainfall from December until the end of February. In the months when we performed our experiment, there was no rainfall, as is typical during this time of year (*Mungai et al., 2016*). The pigeon pea in our study region is a rain-fed crop and is not irrigated or watered in any way, especially as pigeon pea is considered drought-resistant (*Odeny, 2007*). All field activities, including land preparation and weeding, were managed traditionally by hand. Farmers did not apply any chemicals such as herbicides and pesticides on their fields.

Fields ranged from 166 m$^2$ to 577 m$^2$ in size, with mean field size being 332 m$^2$. This is representative of field sizes of such a crop in the study region, where the average smallholder total farm size ranges from only 0.5 to 1.4 hectares (*FAO, 2018*). Field size did not correlate significantly with the proportion of semi-natural area ($F_{2,6} = 2.08$, $R^2 = 0.21$, $p = 0.683$) nor with the proportion of agricultural area ($F_{2,6} = 2.08$, $R^2 = 0.21$, $p = 0.088$) in the 1 km radius surrounding our fields. Field margins were vegetated with non-flowering weeds, grass or shrubland. As it was the dry season during data collection, there were no flower margins on the fields. The surrounding agricultural fields where mostly empty, as the main staples in Malawi, such as maize, was already harvested by this time in the season. Surrounding semi-natural habitat was mainly composed of shrubland and forest. Generally, these are not actively managed but may to some extent be exposed to exploitation by people due to economic activities such as collection of firewood and grazing of livestock.

We aimed to choose sites which were at a distance of at least 2 km from each other. However, one site was found to have too large an overlap with two others within a 1 km radius, with the center of this field being 883 and 885 m away from the center of the nearest and second-nearest site, respectively. Therefore, this site was subsequently dropped from any landscape analyses (Fig. 1). The remaining fields were located within two non-correlating gradients of semi-natural habitat (ranging from 2% to 32%), and agricultural area (ranging from 25% to 75%) within a 1 km radius surrounding the fields

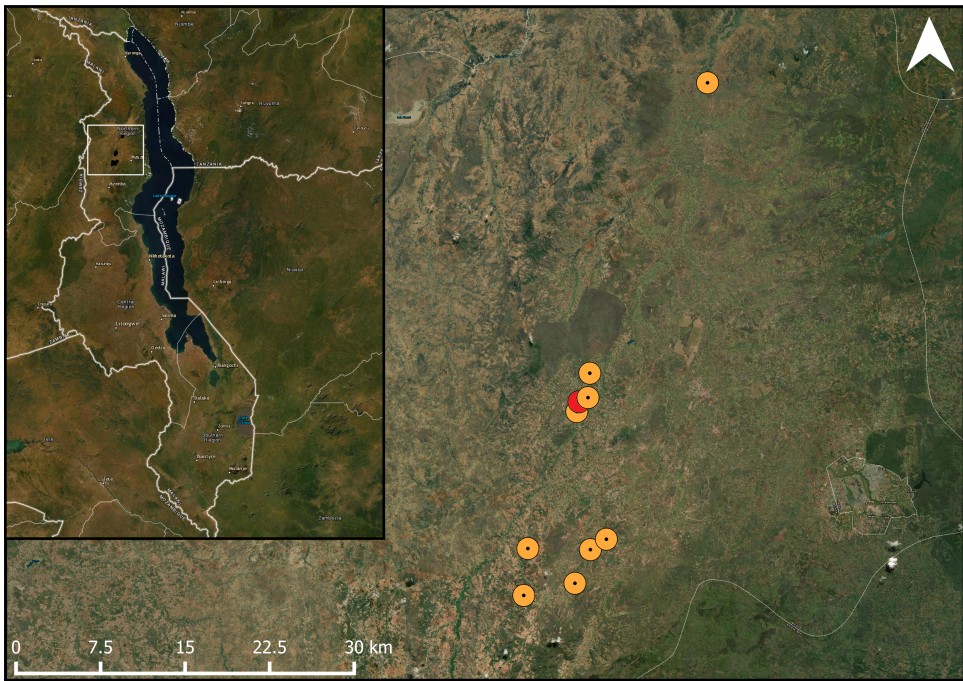

**Figure 1 Studied pigeon pea fields in the landscape.** Map showing the location of the study fields within its one-kilometre buffer within the study area. The study area marked in red had too much overlap within the one-kilometre radius with the adjacent fields. ESRI Satellite is used as a basemap (ArcGIS Pro 2.6; Esri, Redlands, CA, USA).

($F_{1,7} = 0.56$, $R^2 = 0.07$, $p = 0.480$). The 1 km radius was chosen, because we wanted our sites to be independent from each other and prevent spatial autocorrelation. Additionally, since bees are central place foragers, and their foraging ranges are limited, most bees would be sensitive to landscape differences within the 1 km radius (*Steffan-Dewenter et al., 2002*; *Zurbuchen et al., 2010*). Other habitats in our study area included some built-up areas (such as buildings and roads) and bare rock (mostly on hilltops). Although honeybees are native to the area (*Requier et al., 2019*), we found no honeybee hives in any of the fields across our study area. Moreover, none of the farmers we worked with kept honeybees on any of their fields. In our study area, honeybees are rarely actively placed in fields by farmers, but rather encouraged to nest nearby by placing traditional beehives near fields where they may be passively colonized by a honeybee colony (Appendix 2) (*Requier et al., 2019*). To our knowledge, there were no such hives placed near any of our study fields.

## Landscape analysis

For the land use and land cover classification, we acquired three cloud-free Sentinel-2 satellite images from 2019 from the Copernicus Open Access Hub (https://scihub. copernicus.eu/dhus/#/home). One image was taken on November 8th 2019, and two are from November 15th 2019, which is in the late dry season in Malawi. Though this is not the same time as our field study period, the images still show the general land use and land

cover in our study area and we assume this has not changed significantly between May and November.

The methodology for land cover analysis involved images pre-processing, supervised maximum likelihood classification, post-classification, and accuracy assessment (Appendix 3). First, we pre-processed the Sentinel-2 images, which consisted of the atmospheric correction, image resampling, layer stacking, seamless mosaic, and image subsetting. Then, we conducted Maximum Likelihood classification using training samples generated from fieldwork and Google Earth. The classification includes six classes, which are: (1) bare land/road; (2) shrubland; (3) agricultural land; (4) water/riverbed; (5) settlement; (6) trees/forest. However, in this study, we only used classification (2) shrubland and (6) trees/forest together which we defined as semi-natural habitat (SNH) and (3) agricultural area. Finally, we performed post-classification and accuracy assessment. The overall accuracy of the classification is 85.1%, with a Kappa Coefficient of 0.817. We used tabulate area tools in ArcGIS to get the area and proportion of the types of land use and land cover for all buffer zones of each site (ArcGIS Pro 2.6; Esri, Redlands, CA, USA).

## Bee and beetle collection

To assess bee and blister beetle densities, we walked straight 15 m transects for 10 min across the most flower rich area of the field on three separate dates, resulting in 30-minute total sampling time per field. One round of transects at each field was finished before a new round was started, resulting in a pause of about a week between individual sampling dates at each field. All flower-visiting bees one-meter left and right of the investigator were captured with an insect net for subsequent identification. All blister beetles observed one meter left and right of the investigator were counted, taking care not to count the same individual twice. To obtain a proxy for planting density, we counted the number of pigeon pea shrubs across each transect. Planting density across transects did not correlate with the proportion of agricultural area ($p = 0.786$) or semi-natural habitat ($p = 0.338$) ($F_{2,7} = 0.53$, $R^2 = 0.13$, $p = 0.610$). Transects were walked between the 8th and 29th of May 2019, between 8:00 and 16:00, when weather conditions were not windy (Beaufort scale < 3) or too cloudy (<80% cloud cover), and thus deemed favorable for bee foraging. It did not rain, and temperatures ranged from 18 °C to 25 °C throughout our sampling period. Visitation order on a specific day was randomized so that each field was visited during a different time of the day during consecutive transects.

## Bee identification

Captured bees were identified to genus or subgenus level with the guides from *Michener (2007)* and *Connal, Kuhlmann & Pauly (2010)* and grouped by (morpho)species. Captured bees are stored at the Biocentre, University of Würzburg, Germany.

## Flower exclusion and fruit set data

At each site, we marked 15 pigeon pea plants in a continuous line from the edge of the field inwards. On each plant, we tagged one cluster of flowers as the open control. This cluster was accessible to all visitors, both pollinators and herbivores. On the same plant, we then paired this tagged cluster with another cluster of flowers to which all visitors (pollinators as

well as herbivores) were excluded using a $9 \times 12$ cm organza bag. The number of flowers in the tagged and bagged clusters where counted. The bags had a mesh size of 0.6 mm—small enough to exclude any insect. Although exclusion of pollinators and pests in different treatments would have been ideal, this could not be done, since the beetles feed on the flowers, during the same time that pollinators are visiting them. Our hypothesis was that if the fields experience high pest pressure, bagged clusters will perform better, as they are protected from herbivory. On the other hand, we assumed that in fields where there is a large amount of ambient pollination and low flower damage by herbivores, the open clusters would outperform the bagged ones. In fields with little ambient pollination, or where the benefits of pollination are cancelled out by pest damage, open and bagged flowers would perform similarly. Plants were tagged and bagged upon the first visit to the field before the flowers had opened and we removed the bags when all the flowers in the cluster had finished blooming, which took approximately two weeks. After removing the bags, the pigeon pea pods were left to mature in the field.

Fruit set data was collected from the 3rd of July to the 2nd of August 2019. To assess fruit set (the proportion of flowers turning into pods) as a measure of pollinator effectiveness, we counted the number of flowers that were originally present on the tagged clusters, and then counted the number of pods formed in the same clusters. The number of pods formed divided by the number of flowers was taken as a measure of fruit set per cluster. In one field, damage by cattle grazing destroyed the tagged plants and we were unable to collect data on fruit set there.

## Blister beetle damage assessment

To get a measure of the proportion of flowers damaged by blister beetle herbivory/florivory, we assessed flower damage three times at eight of the sites and twice at two of the sites. We used the open cluster of the 15 pigeon pea shrubs we tagged in each field for this. We counted the number of flowers per cluster and the number of these flowers that showed signs of chewing herbivory typical of blister beetles. With this data we calculated the proportion of flowers damaged by blister beetles.

## Data analysis

To test whether landscape composition affected bee and blister beetle abundance, we summed the number of individuals across all three transects. For bee richness, we used the cumulative bee richness across dates per field. We first tested if bee abundance, bee richness and blister beetle abundance were independent of planting density across transects or field size (Appendix 4). We then tested how the proportion of semi-natural habitat and agricultural area within the 1 km radius affected bee and blister beetle abundance using a linear regression. To test whether landscape composition affected bee richness, we used the bee richness at each site and again tested this against the landscape variables using a linear regression. To test to what extent our patterns where driven by the presence of honeybees (*Apis mellifera* L.), the most abundant pollinator in our system, we tested bee abundance against the two landscape variables including and excluding honeybees from the analysis.

To test if landscape variables affected blister beetle damage in our fields, we calculated for each of the 15 plants the mean proportion of flowers damaged by herbivory across
the flowering season from the three dates. Since our data were zero-inflated (no flowers damaged), we used a negative-binomial mixed model using the 'glmer.nb' call from the package 'lme4' (*Bates et al., 2019*). We tested the mean proportion of flowers damaged against the proportion of semi-natural habitat and agricultural area. Since we had repeated measures within fields, we used field as a random factor in this model.

To test whether landscapes affected the differences in fruit set between bagged and open clusters, we calculated the proportion of flowers that set fruit for each cluster. Then, we subtracted the proportion of fruit set of the bagged cluster from that of the open cluster. Again, using the package 'lme4' (*Bates et al., 2019*), we calculated mixed effects models testing the difference in fruit set against the two landscape variables, using field as a random factor to account for repeated measures per field. In this analysis, we had to exclude 31 out of 120 plants due tampering or missing tags.

Finally, to calculate the effect of bee visitation and beetle damage on fruit set difference between the bagged and open clusters, we used mixed models. To do this, we took the total number of bees recorded at each site, and divided this by the total number of pigeon pea shrubs across our transects. This gives us bee density per crop plant, which we used as a proxy for bee visitor density per pigeon pea shrub. We then calculated the effect of bee density, bee richness and the proportion of flowers damaged per plant on the difference in fruit set between the paired clusters, using field as a random factor to account for nestedness. We chose to use flower damage, rather than beetle abundance or density in this model because we deemed it a more concrete representation of the pest pressure the plants experienced, though blister beetle abundance and the proportion of damage was correlated ($F_{1,418} = 4.88$, $R^2 = 0.01$, $p = 0.028$). In this analysis, we had to exclude 34 out of 135 pigeon pea shrubs due to tampering or missing tags.

All models were tested for and met the assumptions of distributions, normality (of residuals) and heteroscedasticity. All statistical analyses were performed in R version 4.0.1 (*R Core Team, 2020*).

## RESULTS

### Landscape effects on bee abundance, bee richness, and blister beetle abundance

In total, we observed 84 bees of 13 species (Appendix 5) and 127 blister beetles across the five hours of transects during our study period. The proportion of semi-natural habitat within a 1 km radius of the fields did not affect bee abundance ($F_{2,6} = 5.53$, $R^2 = 0.65$, $p = 0.775$) (Fig. 2A) or richness ($F_{2,6} = 1.38$, $R^2 = 0.32$, $p = 0.203$) (Fig. 2C), and neither did it affect blister beetle abundance ($F_{2,6} = 1.58$, $R^2 = 0.35$, $p = 0.538$) (Fig. 2E). The proportion of agricultural area positively affected bee abundance ($F_{2,6} = 5.53$, $R^2 = 0.65$, $p = 0.0209$) (Fig. 2B), though this pattern was primarily driven by higher honeybee densities at high-agricultural area sites, as solitary bees alone did not respond significantly to landscape factors. The pattern was additive, as honeybee densities alone did also not show significant patterns, and it was just the analysis with honeybees and solitary bees together that showed a result (Appendix 6). However, agricultural area did not affect bee
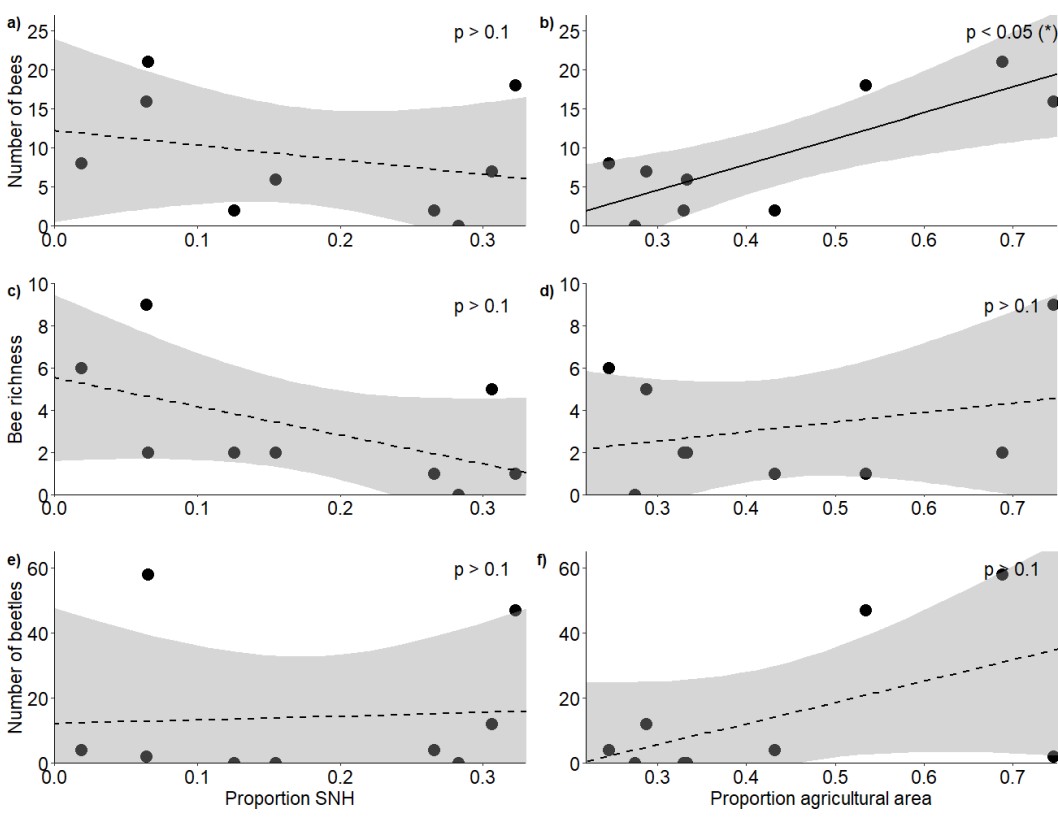

**Figure 2** **Response of bees and blister beetles to landscape variables.** Relationship (±95% CI) between the proportion of semi-natural habitat in the surrounding landscape and (A) bee abundance, (C) bee richness and (E) blister beetle abundance, as well as the relationship between the proportion of agricultural area and (B) bee abundance, (D) bee richness and (F) blister beetle abundance.

**Table 1** **Model summaries of animal responses to landscape composition.** Model summary of linear models assessing bee and blister beetle responses to landscape composition (proportion semi-natural habitat and proportion agricultural area) ($n = 9$).

| Response | F-statistic (2,6) | Multiple R² | p-value | Predictor | t-value | p-value |
|---|---|---|---|---|---|---|
| Bee abundance | 5.53 | 0.65 | 0.043 | SNH | −0.30 | 0.775 |
| | | | | Agricultural area | 3.11 | **0.021** (*) |
| Bee richness | 1.38 | 0.32 | 0.321 | SNH | −1.43 | 0.203 |
| | | | | Agricultural area | 0.43 | 0.683 |
| Blister beetle abundance | 1.58 | 0.35 | 0.281 | SNH | 0.65 | 0.538 |
| | | | | Agricultural area | 1.77 | 0.127 |

richness ($F_{2,6} = 1.38$, $R^2 = 0.32$, $p = 0.683$) (Fig. 2D) nor blister beetle abundance in the fields ($F_{2,6} = 1.58$, $R^2 = 0.35$, $p = 0.127$) (Fig. 2F) (Table 1).

## Landscape effects on blister beetle damage and fruit set difference

The proportion of flower damage ranged from zero to 0.36, with a mean of 0.06. There was no effect of the proportion of semi-natural habitat nor agricultural area on the proportion of flowers damaged by blister beetles on the tagged open clusters (Appendix 7). The
**Table 2 Summaries of the models assessing the landscape composition on the proportion of damaged flowers and fruit set difference.** Summary of the linear mixed models assessing the effect of landscape composition (proportion semi-natural habitat and proportion agricultural area) on the proportion of damaged flowers and the difference in fruit set between the open and the bagged treatment.

| Response | Total number of observations | Number of groups (n) | Predictor | z-value | p-value |
|---|---|---|---|---|---|
| **Proportion of damaged flowers** | 135 | 9 | **SNH** | 0.05 | 0.960 |
| | | | **Agricultural area** | −0.78 | 0.439 |
| | *t-value* | *p-value* | | | |
| **Difference in fruit set** (open – bagged) | 89 | 8 | **SNH** | −1.83 | 0.125 |
| | | | **Agricultural area** | −0.19 | 0.853 |

**Table 3 Model assessing the effect of bees and blister beetles on fruit set difference.** Summaries of the linear mixed model assessing the effect of bee density, bee richness and proportion of damaged flowers on the difference in fruit set between bagged and open flower clusters.

| Response | Total number of observations | Number of groups (n) | Predictor | t-value | p-value |
|---|---|---|---|---|---|
| **Fruit set differences** (open - bagged) | 101 | 9 | **Bee density** | 0.36 | 0.734 |
| | | | **Bee richness** | 1.45 | 0.196 |
| | | | **Proportion of flowers damaged** | 1.923 | 0.058 |

number of open clusters that a higher proportion of damage than 0.05 varied from 2 to 12 clusters per site, but this did not correlate to either proportion of semi-natural habitat ($F_{2,6} = 0.01$, $R^2 = 0.001$, $p = 0.923$) or agricultural area ($F_{2,6} = 0.01$, $R^2 = 0.001$, $p = 0.998$). The proportion of fruit set on open clusters ranged from 0 (none of the flowers set fruit) to 1 (all flowers set fruit) with a mean fruit set proportion of 0.37. The proportion of fruit set on bagged clusters also ranged from 0 to 1 with a mean fruit set proportion of 0.26. Fruit set difference (open-bagged) ranged from −1 to 1 and had a mean of 0.11. There was no effect of the landscape variables on the fruit set difference between the open and bagged flower clusters (Table 2) (Appendix 7).

### Effects of bees and blister beetle damage on fruit set difference

We observed no effects of bee density or bee richness nor of blister beetle damage on the proportional difference in fruit set (Table 3) (Appendix 8).

## DISCUSSION

In our study, we aimed to investigate how differences in landscape composition may drive ecosystem services and disservices on smallholder farms in the tropics. We find that increasing agricultural area surrounding our crop increases the abundance of bees, driven primarily by an increase in honeybees, on our studied fields. This seems in contrast with most studies that indicate that increasing semi-natural habitat in the surrounding area increases pollinator abundance in crop fields (*Kennedy et al., 2013*). Another study on pigeon pea, conducted in Kenya, also showed that fields located closer to semi-natural habitat also had a lower abundance of pollinators (*Otieno et al., 2011*), indicating that such a pattern may be more common in the African seasonal tropics. In our system, pigeon pea flowers during May, which is well into the dry season in our study system (*Mungai*

*et al., 2016*). In general, bee abundance in our study system was low, which is expected in the seasonal tropics where the peak in insect activity is usually on the onset of the wet season, which in our study area would be around December (*Kishimoto-Yamada & Itioka, 2015*). Since abundances were mostly driven by honeybees, it can be assumed this could be due to the larger number of colonies nesting in agricultural areas compared to areas with less agricultural area. Relative to solitary bees, honeybees also have a larger foraging range, which means they may be more successful in finding resource-rich flowering fields in a resource-poor environment over larger distances from their nests. Additionally, they recruit colony mates to forage there, which is not the case for solitary bees, which do not live in colonies and have more limited foraging ranges (*Steffan-Dewenter & Kuhn, 2003*; *Zurbuchen et al., 2010*). Considering the resource scarcity during our study period, it is reasonable to observe higher densities of bees in agricultural areas, where there are still some flowering crops providing resources to bees, which would be almost absent in semi-natural areas during this time of year.

Aside from one site, most of our sites showed similar performance between bagged and open clusters. In our study, damage by blister beetles did not predict differences in fruit set between bagged and open flower clusters. Though we do not rule out that blister beetles contribute to losses in fruit set, our data suggests blister beetles are not as significant a pest on pigeon pea in our study area as commonly believed. Farmers often state blister beetles as a significant constraint to growing pigeon pea in our study area, perhaps because they are conspicuous (*Mungai et al., 2016*). In our study, we used existing pigeon pea fields, and did not plant the fields specifically. Since no pigeon pea fields existed in areas where farmers did not grow pigeon pea due to extensive pest damage, we did not investigate those areas where blister beetle densities are perceived to be highest. It could be that the contrast between our sites is not large enough to observe possible differences since these higher extremes are not included. This could contribute to the fact that we did not observe any differences in blister beetle abundance on pigeon pea fields and resulting flower damage within the scope of our study, and pest damage was similar across sites.

Increasing bee densities did not improve the fruit set of open-pollinated flower clusters compared to bagged clusters. This is in contrast with many studies showing improved agricultural production with increased flower visitation, particularly on small farms like those in our study system (*Garibaldi et al., 2016*), and also on an earlier study on pigeon pea (*Otieno et al., 2011*). Additionally, we did not find an effect of bee richness on fruit set, which is also not consistent with other studies on pollinator dependent crops (*Garibaldi et al., 2016*; *Dainese et al., 2019*). In our study, higher bee abundances were mostly driven by increased honeybee abundance. Previous studies indicate that honeybee visitation often does not benefit crop yield (*Garibaldi et al., 2013*), which could explain the lack of increased fruit set in our sites with higher bee abundance. Numerous studies have shown that pollination and pest damage may interactively shape crop yields (*Lundin et al., 2013*; *Bartomeus, Gagic & Bommarco, 2015*). In our study, such effects may also be at play, but we cannot distinguish them since we were unable to test the interactive effects due to low sample size.

In our study area, honeybees were the most abundant bees, and therefore important in driving higher bee abundances in agricultural areas. In contrast to temperate systems, in Africa, up to 90% of honeybee colonies occur in the wild, and honeybee keeping as a practice is still underdeveloped and small-scale, with no impact of humans on breeding (*Requier et al., 2019*). Therefore, like both social and solitary wild bees in temperate systems, honeybees in our system have conservation value as a part of the local bee biodiversity (*Dietemann, Pirk & Crewe, 2009*). Though bee visitation did not directly benefit fruit set of this particular crop, the fact that a crop flowers during this time of year may still be important, as this could provide an important flower resource, particularly for social bees, that are still active during this season of scarcity in our study system. If it helps individual bees and honeybee colonies to survive this time of the year, it may benefit farmers on the long run if these pollinator populations are conserved until the next growing season when the farmers may be growing early flowering crops that are more strongly pollinator dependent.

## CONCLUSIONS

Many studies show the effect of landscape composition on the abundance and richness of pollinators and pests, and that particularly in the case of pests, these patterns are not always consistent (*Kennedy et al., 2013*; *Karp et al., 2018*). Our study shows that these patterns may be quite different in tropical smallholder agriculture compared to better studied tropical agroforestry and temperate annual cropping systems. Differences in patterns in comparison to temperate systems highlight the necessity to study different climatic and growing contexts better. Our study indicates that late-flowering crops provide an important floral resource during a scarce period in the seasonal tropics and are therefore an important component in sustainable agriculture in these parts of the world.

## ACKNOWLEDGEMENTS

We are grateful to Innocent Mhoni, Pressings Moyo, Tapiwa Mkandawire and Mwapi Mkandawire for their assistance in the field. We are also grateful to the leadership of the Soils Food and Healthy Communities (SFHC), Esther Lupafya, Laifofo Dakishoni and Lizzie Shumba for all their help streamlining all practical aspects of performing the study in Malawi. We are grateful to Aaron Iverson for his input on the data and practical advice during the field season. We also owe Jinfei Wang and Daniel Kpienbaareh our thanks for the teamwork required to complete the landscape analysis. We also thank the reviewers for their contribution to the improvement of the manuscript. Finally, we owe our gratitude to all the farmers who allowed us to perform our experiments in their fields.

### Funding

This research was funded through the 2017–2018 Belmont Forum and BiodivERsA joint call for research proposals, under the BiodivScen ERA-Net COFUND programme, and

with the funding from Natural Sciences and Engineering Research Council (NSERC Grant# 523660-2018). Cassandra Vogel is funded through the German Federal Ministry of Education and Research. This publication was supported by the Open Access Publication Fund of the University of Würzburg. The funders had no role in study design, data collection and analysis, decision to publish, or preparation of the manuscript.

**Grant Disclosures**

The following grant information was disclosed by the authors:
Belmont Forum and BiodivERsA.
Natural Sciences and Engineering Research Council: 523660-2018.
German Federal Ministry of Education and Research.

**Competing Interests**

The authors declare there are no competing interests.

**Author Contributions**

- Cassandra Vogel conceived and designed the experiments, performed the experiments, analyzed the data, prepared figures and/or tables, authored or reviewed drafts of the paper, and approved the final draft.
- Timothy L. Chunga performed the experiments, authored or reviewed drafts of the paper, and approved the final draft.
- Xiaoxuan Sun analyzed the data, authored or reviewed drafts of the paper, and approved the final draft.
- Katja Poveda and Ingolf Steffan-Dewenter conceived and designed the experiments, authored or reviewed drafts of the paper, and approved the final draft.

**Field Study Permissions**

The following information was supplied relating to field study approvals (i.e., approving body and any reference numbers):

We were granted verbal permission for conducting the research on each of the farmer's private fields. Their names are:

V2: Isobel Lubanda
V4: Adams Tembo
V25: Mercillina Tembo
V26: Ireen Mhoni
V27: Simon Chitaya
V28: Jacob Mvula
V29: Jane Salanda
V30: Lyna Njunga
V31: Goodson Moyo
V32: Moles Thupa

The farmers are not represented by a company or a farming cooperation, but were in contact with the authors through the SFHC (Soils, Food and Healthy Communities) organisation. We have no form of written permission for the conduction of the research.
## Data Availability

Raw data, including the number of flowers and the number of damaged flowers, is available in the Supplemental Files.

## Supplemental Information

Supplemental information for this article can be found online at http://dx.doi.org/10.7717/peerj.10732#supplemental-information.

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
