# Peer review of "Higher bee abundance, but not pest abundance, in landscapes with more agriculture on a late-flowering legume crop in tropical smallholder farms"

_PeerJ, doi:10.7717/peerj.10732_

## Round 0.1 · original submission · Major Revisions

Dear Dr. Vogel and colleagues:

Thanks for submitting your manuscript to PeerJ. I have now received three independent reviews of your work, and as you will see, the reviewers raised some concerns about the research. Despite this, these reviewers are optimistic about your work and the potential impact it will have on research studying honeybee-associated agriculture and landscape ecology. Thus, I encourage you to revise your manuscript, accordingly, taking into account all of the concerns raised by the three reviewers.

Major problems with your study include 10 min surveys (deemed insufficient) and the lack of necessary information about the sites. Please address these in your revision and rebuttal letter. Other concerns over methodology must also be addressed.

There are many specific issues pointed out by the reviewers, and you will need to address all of these and expect a thorough review of your revised manuscript by these same reviewers.

Therefore, I am recommending that you revise your manuscript, accordingly, taking into account all of the issues raised by the reviewers.

I look forward to seeing your revision, and thanks again for submitting your work to PeerJ.

Good luck with your revision,

-joe

·

Basic reporting

1. Basic Reporting
The title should be higher honeybee densities, not bee densities.
Abstract:
L26: Apiformes should not be italicised – only the species binomial name is italicised.
L30: Just state bees, rather than pollinating bees. Not all bees are pollinators, and this study did not test their pollinating ability.
L36: How long were the transect, what length of time did you walk them? How many people walked them? Were bees captured?
Results should be written in past tense, for example: we did not, rather than we do not
L43: Remove “we see that”
Discussion section: Again, this should be written in past tense.
Did you quantify resources in natural areas? Is this an assumption that alternative foraging resources were scares there?
Introduction:
L69: Insert ‘a’ before ‘recent meta-analysis’
L79: large rather than larger
L83: I would say commercially important, not ‘interesting’
L85: Reference required for Africa’s food security issues
L86: Pesticides however don’t necessarily reduce crop losses – I think this should be acknowledged
L104: Apiformes shouldn’t be italicised.
Results: I am confused as to why you report bee abundance and species richness for landscape variables, but then bee density and accumulative bee species richness when looking at fruit set. Can you please be consistent with the metrics you use in reporting results?
Discussion:
L253: it didn’t increase the abundance of pollinators: rather, it increased the abundance of honeybees.
Whether this is due to a beneficial effect of agricultural crops on honeybees is also uncertain. As you rightfully note in L263, surrounding agricultural land had managed honeybees which were the source of the increased honeybees observed on nearby pea fields. I would advise seeing if you could quantify honeybee colony densities in the landscape.
L304: I don’t think you can justify this statement. Firstly, you found increasing honeybees did not increase fruit set. Secondly, there is the possibility honeybees are excluding native bees – you have a small sample size, but I would recommend seeing if there is a negative correlation between honeybees and the wild solitary bees. It may be that if honeybees were absent, there would be more wild solitary bees, who may be better pollinators of pea plants (which has been found in diverse pea crop species and wild crop species: e.g. Aouar-Sadli, M., Louadi, K., & Doum, S. E. (2008). Pollination of the broad bean (Vicia faba L. var. major)(Fabaceae) by wild bees and honey bees (Hymenoptera: Apoidea) and its impact on the seed production in the Tizi-Ouzou area (Algeria). African Journal of Agricultural Research, 3(4), 266-272; Aouar-Sadli, M., Louadi, K., & Doum, S. E. (2008). Pollination of the broad bean (Vicia faba L. var. major)(Fabaceae) by wild bees and honey bees (Hymenoptera: Apoidea) and its impact on the seed production in the Tizi-Ouzou area (Algeria). African Journal of Agricultural Research, 3(4), 266-272; Gross, C. L. (2001). The effect of introduced honeybees on native bee visitation and fruit-set in Dillwynia juniperina (Fabaceae) in a fragmented ecosystem. Biological conservation, 102(1), 89-95.)
Conclusion: you did not measure landscape complexity per se, so I would rephrase this.
References:
Inconsistent formatting. Please fix.

Experimental design

2. Experimental design
Methods:
Please provide information on how large each field was. Also, were peas the only flowering plants in the fields or were wildflowers/flowering weeds also present?
Were fields all sampled on the same day?
There is a big difference between many bee assemblages being active at 08:00 vs. 16:00, with resources being depleted across the day. Did you randomise when each field was surveyed over the day?
L132: How were they captured? I find it unlikely that ALL were captured – were bees that were observed but not captured counted? How many people conducted the sampling? Was it the same investigator each survey? This is important because if not you may need to account for inter-observer error.
L136: Did you investigate if pea planting density influenced bees or beetles? You have the data, so I recommend looking at this, and ensuring flower density if significant didn’t co-vary with landscape variables.
L142: What were these morphospecies groupings? Who identified the bees? Where were they deposited (which institution/museum?)
L115: What variety was this please?
Linear regression is inappropriate – as you revisited the same field sites mixed effect models with site as a random factor should be used.
L214: this is proportion data, and so a glmer with the binomial family should be used.
L227: how were these assumptions tested? If the assumptions were not met, what did you do?
Given the dominance of honeybees and their different life-history to native bees, I would recommend doing the statistical analyses separately for honeybees vs. native bees regarding abundance.
Were managed honeybees kept on any of the farms?
Were the pea plants field treated with herbicides or insecticides?
I strongly believe 10 mins over 15 m is inadequate sampling effort. I recommend conducting the study over another year and sampling for a longer duration.

Validity of the findings

3. Validity of the findings

The limited sampling duration limits the validity of the findings, and different statistical tests are required (include site as a random effect to account for re-visits).
More information on the methodology is required to be able to judge the validity of the results.

Results: Please also conduct analyses on the influence of proportion of natural habitat on native bee abundance and honeybee abundance separately, and the relationship between honeybee abundance and proportion of agricultural habitat. I acknowledge that at present the sample size is small, hence I recommend conducting another year of surveys and increasing the sampling effort.

Additional comments

Thank you for you research into what you point out, is an understudied region. Although I believe this research has great potential, there are a number of factors that need to be addressed before this manuscript is of publishable quality. I hope you find my comments and feedback useful.

This article contributes to studies outside of northern hemisphere temperate ecosystems, which are certainly understudied, and I applaud the authors for undertaking this understudied topic of research.
Please have a taxonomist identify your specimens, and provide a list of the species, and where they are deposited.

Reviewer 2 ·

Basic reporting

The authors report the results from a landscape study on pollinators and pests in pigeon pea in Malawi. The manuscript is well-structured. The language can be more concise (see many detailed comments on this provided below). Some more background information would be helpful, if it exists (see detailed comments on the introduction below).

Experimental design

The research questions are well defined and the research fills an identified knowledge gap. The research design and execution is sound. The bagging treatment is mixing effects of pollinators and florivorous pests on the crop, but this is acknowledged. Sample sizes are limited (number of sites, number of insects collected), but sufficient for statistical analyses. The methodology should be clarified in some cases (see detailed comments provided).

Validity of the findings

See below a few minor comments on the underlying data files provided.

Additional comments

Detailed comments

Abstract
L36. blister beetles
L41. Consider to delete “We show that” (it is clear from the heading “Results” that these are your results.
L43. Delete ”We see that”. blister beetle
L44-45. We did not
L49. explaining -> which could explain
L50. “high proportions of SNH” Also, use, or avoid using, the abbreviation SNH in some consistent manner (compare with L28, 29). Check throughout manuscript. Introduce the abbreviation once in the abstract and once in the main text if you are using it.

Introduction
L59. world’s
L60. Suggest to delete “real”
L72. It would be helpful to clarify a bit what you mean by “different contexts”
L76-77. Clarify what is the difference between this argument and that SNH can be a source of pests (mentioned on L75)?
L81. “within the African context”. Consider to write “in Africa” instead.
L88. Specify the regions or reformulate “in African regions” to “in Africa”.
L90. Italicize the scientific name.
L91. due to its
L93. Recommend to use pigeon pea (singular, as on L90) or at least not mix singular and plural. Comment applies throughout manuscript (and also for groundnut/s).
L93-95. It would be helpful to provide if there is more information about this pest. Is it one or several species? How do they damage yield / do they eat the reproductive parts of the flowers?
L96-97. Clarify what are the pests causing these yield losses.
L98-100. It is unclear to me from this statement what proportion of pigeon pea yield that is pollinator dependent. Bees provide 20-90% of the cross pollination, but unclear what vectors provide the rest (other insects?) and the degree of selfing of the crop (or is it fully cross-pollinated?).
L102. Consider to delete “Considering all of the above,”
L105. Consider if you need to modify this statement here, and also in the abstract because you are not using fruit set directly in analyses, but rather always the difference in fruit set between open and bagged.

Materials and Methods
L113. Provide the scientific name for groundnut.
L116. Suggest to delete “very”, L120 “quite” etc throughout as these words to not add any objective meaning.
L121-122. Specify the overlap by providing the distances to the two nearby fields. The overlap does not seem large to me based on the map.
L123-126. Some justification for choosing the 1 km radius would be useful.
L137. It is unclear to me what you mean by counting pigeon peas (plant individuals? pea numbers?) across the transect, and how this could serve as a proxy for planting density. Also unclear why planting density is an important covariate. It seems flower density could have been an important covariate though when you count or collect flower-visiting insects.
L138. Three times. Repetition from above.
L139. It would be good to specify how you define weather that is favorable for bees.
L156. we removed
L160. Delete “actually”.
L161. “three times at eight”
L162. How many clusters were assessed each visit at each site and how do you separate blister beetle damage from damage caused by other pests?
L167. How come you focus on pods produced per flower as a measure of pollination success only? What about peas per pod, or pea weight?
L168. It would be helpful to mention that flowers were counted when they were bagged in the “Flower exclusion” paragraph. You can consider to merge this paragraph with that one.
L172-173. Clarify the proportion missing data.
L175. I recommend to have this paragraph earlier in the methods, after “Study area and field selection” (where you talk about the landscape composition around your fields).
L181. involved (methods are normally written in past tense, check this throughout).
L225. Suggest to delete “actually”.
L227. I think it would be good if you clarify a bit what assumptions you tested for.

Results
L231. Delete “different”.
There are only two figures and three tables in the main text, so I think it would be possible to include some (or all) of the supplementary tables and figures in the main text.

Discussion
L260, 271, 272, 286, 302. Delete “rather”, “completely”, “seems to”, “seem to”, “rather”,
L273-284. Some condensing of this text is possible.
L313. Clarify what you mean with “complex landscape effects”.
L317. contexts
L317-320. I do not think you need triple expressions of uncertainty, especially as this is your concluding sentence. “indicates” in the beginning of the text is enough to express uncertainty across the full line of thought. You can remove “could” and “might” from this sentence.

Appendix 1
mosaic (typo). Check for consistent use of capital letters.

Figure 2, legend
“between between” (word duplicated by mistake). Relationship between A and B, not relationship between A on B.

Table 2.
Provide p-value decimals in some consistent manner. Unclear why it says “(Log)” after “Proportion of damaged flowers”. I did not see this being explained anywhere.

Data files
Proportion damage is termed “percent.damage” in the Excel file, but the numbers are proportions.
Please check that you use percent and proportion correctly here and throughout the manuscript.

Was does the comment “Empty pod” mean in fruit set data? If a pod is formed but contain no peas should it then contribute to pod set?

Was does the variable “Overlap.100m” mean in the landscape data?

·

Basic reporting

L. 90: 'Cajanus cajan' should be written in italics.
L. 106: 'SNH' is mentioned here, but not yet explained previously.

L. 115: Here it is mentioned that the peak of the bloom is in May. I think this might be a good place to mention the duration of the bloom here as well.

Could you please present Table 3 in the same format as Table 2? A similar type of results are presented in both tables, yet the way of presenting makes Table 2 much more accessible. Furthermore, it increases the consistency of the manuscript, and hence the overall readability.

Experimental design

Could you please mention the size of the studied pigeon pea fields? I think this contributes to the reader's overall conceptualization of the study.

Could you please mention the mesh size of the used organza bags?

L. 242 – 245: This part of the results concerns the ‘Landscape effects on blister beetle damage and fruit set difference’ as written in the subtitle. Yet I believe the effects on fruit set are not reported in the text. Effects of the landscape on ‘blister beetle damage’ and ‘the proportion of damaged flowers on the tagged open clusters’ are mentioned. Yet the effects of the landscape on the difference in fruit set between open and bagged flower clusters is not mentioned in the text.

Validity of the findings

No comment.

Additional comments

Dear authors,

You performed an interesting study, which you performed and discussed carefully. All this work is presented in a very well-written manuscript. I only have a few minor remarks.

---

## Round 0.2 · Minor Revisions

Dear Dr. Vogel and colleagues:

Thanks for revising your manuscript. The reviewers are very satisfied with your revision (as am I). Great! However, there are a few additional concerns to address per reviewer 2. Please attend to these issues ASAP so we may move towards acceptance of your work.

Best,

-joe

Reviewer 2 ·

Basic reporting

I have re-read the manuscript following the revision. I am satisfied with how the authors have dealt with most of my comments and those of the other reviewers, and I think that the manuscript has improved significantly as a result. I have a few more comments, most of them as a result of the revisions made.

L28. You can remove the acronym SNH from the abstract since you now decided not to use the abbreviation in the abstract.

L30, 32. I think the abstract would become more clear if you replace “visited and unvisited” with bagged and open and bagged flowers (or add bagged and open within parenthesis) (note that you use "bagged and open" on L44).

L39. “of visited flowers” is confusing here. I think you can remove it.

L87-89. Add references for this new sentence, regarding (1) increased pesticide use in Africa, (2) lack of target effects of pesticides, and (3) side-effects human health and the environment.

L101-102. Clarify if you are referring to one ("Hycleus sp. is a common pest") or multiple pest species here ("Hycleus spp. are common pests").

L119. Why “fruit set on open flowers” here for the pollinators (but fruit set difference for pest damage)? As stated previously, you always analyze effects on fruit set difference.

L196, L298, L360. did not, cannot.

Check your manuscript carefully for small mistakes such as mix of capital and non-capital letters (Appendix 2, appendix 3) and double spaces.

Experimental design

L138-140. Provide separate results for the two landscape variables.

L147-148. This is still not clear. According to the map, this one site seems to be more than 885 meters from other sites. Do you mean 885 m from the outer edge of the 1km landscape buffer of another site? Also, was it exactly at the same distance from both its neighbor sites?

L272-273. Does this exclusion of data also applies to the analysis in the previous paragraph?

L286. With the revised information about presence of unmanaged honeybees in the study area, I find this distinction between the two groups “honeybees” and “wild bees” confusing. If I understand correctly honeybees are mostly wild bees in your study area. In either case, use the same group name everywhere (non-honeybees are termed solitary bees in eg Appendix 6)

L298-299. Why the addition of “LM:” here – what does it mean and why is it only used here?

Validity of the findings

L367. Why only solitary bees, and not wild social bees.

Supplements:

I do still not understand all variables in the provided data files. Metadata description (for example, an explanation of each variable name what it means) would be helpful. How do I eg interpret that "Overlap.1000m" > 0 for five of the ten sites? Maybe this is a mistake on my end, and I am just not looking in the correct place, because the authors state that they have “explained what it means in the datafile description provided by PeerJ”. Please clarify.

·

Basic reporting

No comment

Experimental design

No comment

Validity of the findings

No comment

Additional comments

Dear editor and authors,
To my opinion, the authors have carefully revised the manuscript as a great effort to address all my concerns, as well as those of the other reviewers. This has significantly improved the manuscript, and I have no further concerns.

---

## Round 0.3 · accepted · Accept

Dear Dr. Vogel and colleagues:

Thanks for revising your manuscript based on the concerns raised by the reviewer. I now believe that your manuscript is suitable for publication. Congratulations! I look forward to seeing this work in print, and I anticipate it being an important resource for groups studying honeybee-associated agriculture and landscape ecology. Thanks again for choosing PeerJ to publish such important work.

Best,

-joe